# Obstacle Avoidance Strategy for Mobile Robot Based on Monocular Camera

**Thai-Viet Dang** [1],* and **Ngoc-Tam Bui** [2]

1  Department of Mechatronics, School of Mechanical Engineering, Hanoi University of Science and Technology, Hanoi 10000, Vietnam
2  Shibaura Institute of Technology, Saitama 337-8570, Japan
*  Correspondence: viet.dangthai@hust.edu.vn; Tel.: +84-09-8945-8581

**Abstract:** This research paper proposes a real-time obstacle avoidance strategy for mobile robots with a monocular camera. The approach uses a binary semantic segmentation FCN-VGG-16 to extract features from images captured by the monocular camera and estimate the position and distance of obstacles in the robot's environment. Segmented images are used to create the frontal view of a mobile robot. Then, the optimized path planning based on the enhanced A* algorithm with a set of weighted factors, such as collision, path, and smooth cost improves the performance of a mobile robot's path. In addition, a collision-free and smooth obstacle avoidance strategy will be devised by optimizing the cost functions. Lastly, the results of our evaluation show that the approach successfully detects and avoids static and dynamic obstacles in real time with high accuracy, efficiency, and smooth steering with low angle changes. Our approach offers a potential solution for obstacle avoidance in both global and local path planning, addressing the challenges of complex environments while minimizing the need for expensive and complicated sensor systems.

**Keywords:** A* algorithm; computer vision; mobile robot; obstacle avoidance; path planning

## 1. Introduction

The capacity to avoid obstacles is the primary goal of autonomous navigation, which demands precision and efficiency. Thus, autonomous mobile robots (AMRs) must be able to identify environmental barriers and design avoidance maneuvers. Sonars [1], infrared (IR) sensors [2], laser scanners [3], and cameras [4] are only a few examples of the many technologies developed to address the challenge of mobile robot navigation. Sonar devices in various mounting configurations were utilized for many years for obstacle detection due to their great low cost, although suffering from large inaccuracy due to reflection [5]. Due to the repeated teaching process, the processing time is extremely lengthy. Because the path finding method can be developed based on the connection between the destination and target point, the navigation process is not ideal. Laser scanners were shown to be trustworthy and accurate. Nevertheless, they become problematic in uncharted territory [6]. Cameras give rich scene information despite their small footprint, inexpensive cost, and high computing power requirements [6]. The proposed method removed those restrictions because low-cost monocular cameras of sufficient precision are now widely available. Because of their perceptual abilities, AMRs recently saw a rise in popularity as a sensing approach based on vision-based navigation [7]. Recent years saw extensive research into monocular depth estimates using deep learning, and the results are encouraging [8]. Hence, real-time image segmentation-based navigation was effectively completed [9].

In many vision-based applications, such as scene understanding, robotic perception, and picture reduction [10–14], semantic segmentation using deep learning (DL) is a fundamental problem. Minae et al. [10] examined DL-based segmentation models that showed remarkable performance in visual image segmentation tests to address lack of regular datasets for evaluating object segmentation. Then, using a shared dataset, Li et al. [11]

selected the most appropriate strategy from various structures of semantic segmentation. The ISPRS Vaihingen 2D semantic labeling contest was enhanced in [12] to address this issue. With limited samples, the semi-supervised self-learning method maintained semantic segmentation accuracy. Image semantic segmentation with hierarchical feature fusion (ISHF) was presented by Yang et al. [13] to ensure the accuracy of picture segmentation in deep neural networks. Despite ensuring the accuracy of segmentation, there were issues with improving segmentation speed. Fusic et al. [14] proposed a vision sensor-based DL algorithm to classify obstacles and terrain from the evaluation of the obtained image files. However, the navigation based on scene terrain classification was not ultimately demonstrated. Moreover, the vision-based navigation was enhanced by combining it with the sensor system. Gharajeh et al. [15] proposed mobile robot path planning based on the ANFIS technique. However, to save memory resources and fast processing time in the perception, the authors applied binary semantic segmentation, such as moving and restricted regions. The segmentation framework is now more efficient and lightweight without sacrificing quality. There is now both a global and a local path to the path planning process. The performance of AMRs' motion depends on environmental perception [15–23].

Global path planning is only applicable in well-known environments when utilizing well-known techniques, such as the Dijkstra algorithm [16], the A* algorithm [17], and the rapidly exploring random tree (RRT) search method [18]. In close quarters, we employ local path planning strategies, such as the dynamic window approach (DWA) [19], game-theory-based path planning [20], and the ant colony method [21]. Shaher et al. [16] presented a simple, optimal Dijkstra algorithm-based technique for global path planning. Nevertheless, the technique simply restricted nodes with less random access memory (RAM). As a result, the search was essentially inefficient and slow. Eshtehardian et al. [18] developed a rapid RRT* search algorithm combining with B-spline for a smooth trajectory. Although the standard rapid random tree search algorithm entirely handled MOOPs such as smoothness, shortest path, and collision avoidance, the ideal technique required more memory and processing time. Although it converged slowly, [21]'s ant colony algorithm was trustworthy. Based on the Dijkstra algorithm [16], the A* algorithm was a heuristic search tool [17]. In a large and complicated environment, the A* algorithm alone could not be used for path planning. Yonggang et al. [23] suggested an efficient A* approach combining with the three-time Bezier curve to address the concerns of diverse turning points and steering angles. Enhanced A* algorithms are widely utilized for both global and local path planning due to their fast calculation speeds, path optimization, and other advantages.

The authors successfully design multi-scale fully convolutional network-based semantic segmentation for mobile robot navigation using a low resource system. Then, we completely utilize perspective correction on the segmented image to generate the AMR's frontal view of the moving environment, which detects the real-time moving area. In addition, the optimized path planning algorithm is implemented based on the monocular camera. Data processing, thus, calls for adequate performance and speed rate despite constrained means. All three costs work together to guarantee the shortest path, the minimum distance to obstacles, and a smooth trajectory. The structure of the paper consists of the following sections: After the introduction, the binary semantic segmentation based on VGG-16 provides two foundational stages of segmentation network architecture and training process. Next, navigation strategy based on ground plane segmentation describes the foundational stages for constructing an optimal obstacle avoidance navigation strategy. In addition, the approach is strengthened and effectively validated in Experimental Results. The conclusion finishes with a summary and a development of future projects.

## 2. Binary Semantic Segmentation FCN-VGG-16

The suggested binary semantic segmentation based on VGG-16 has two primary components: network architecture and network training. The following two sections elaborate on these two sections (Sections 2.1 and 2.2).

### 2.1. Network Architecture

The authors develop a network based on the FCN [24–27] to accomplish instant pixel-wise labeling while maintaining a decent segmentation outcome. VGG [25] is chosen over AlexNet [26] because of its popularity and more accurate predictions. In contrast to prior work by Shelhamer et al. [27], our binary semantic segmentation based on VGG-16 [7] employs deconvolutional layers at a total of four scales. The authors use different scales to refine their predictions significantly. According to Yang et al. [28], this architecture starts with a low-resolution, rough output prediction and refines it by fusing with prior layers to give both local and global reasoning. VGG-16 is introduced as a convolutional and max pooling layer-based encoder block in this case, as shown in Figure 1. All of the hidden layers rely on linear rectifiers as their activation source. The completed VGG-16 layers are removed from the network. The authors use multiscale fusions to merge features taken from various layers to build the decoder block. The network input is a $96 \times 96 \times 3$ RGB image, and the first scaler's output is 1/32 of the input image's size. The network will then be upsampled to the necessary image size of $96 \times 96 \times 64$ by a classifier transforming the shape to $96 \times 96 \times C$. The number of classes C is used for intending to segment semantically. The number of classes C is set to two because we only wish to mark pixels that belong to the ground and those that do not. Ceilings, doorways, and pillars were not labeled because they are unimportant for the majority of robot navigation purposes, although their labels might be simply added if necessary.

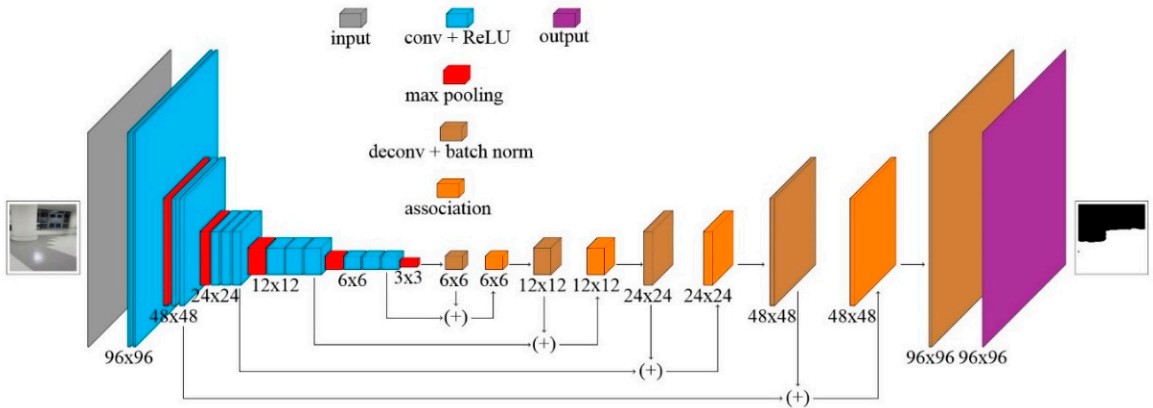

**Figure 1.** Multi-scale fully convolutional network [27].

### 2.2. Network Training

The experiments were executed on a server outfitted with specific software and an operating system. To be precise, the server was equipped with Python 3.11.0 and the TensorFlow 1.4 framework and ran on 64-bit Windows 10 Home English. The server was powered by an Intel(R) Core (TM) I7-8750 h processor, operating at either 2.20 GHz or 2.21 GHz.

$$BCE = -\sum_{i=1}^{C=2} y_i \log(\hat{y}_i) = -y_1 \log(\hat{y}_1) - (1 - y_1)\log(1 - \hat{y}_1) \tag{1}$$

where $\hat{y}_i$ is the class softmax probability; $y_i$ is the prediction's ground truth. The dataset is similar to the source of data in [7]. The corridor environment plays an essential role in AMRs motion, which often motivates the authors to gather corridor-specific training datasets. The forecast of FCN-VGG-16 is depicted in detail in Figure 2 based on Equation (1).

Because of the difficulty of categorizing data, the authors settle on cross-entropy loss. For classification models where the output is a probability value between 0 and 1, a standard performance measure is cross-entropy loss, often known as log loss. While cross-entropy and log loss have subtle differences depending on the setting, they are equal for assessing error rates between 0 and 1 in machine learning. When C = 2, it is referred to as binary cross-entropy or BCE in Equation (1).

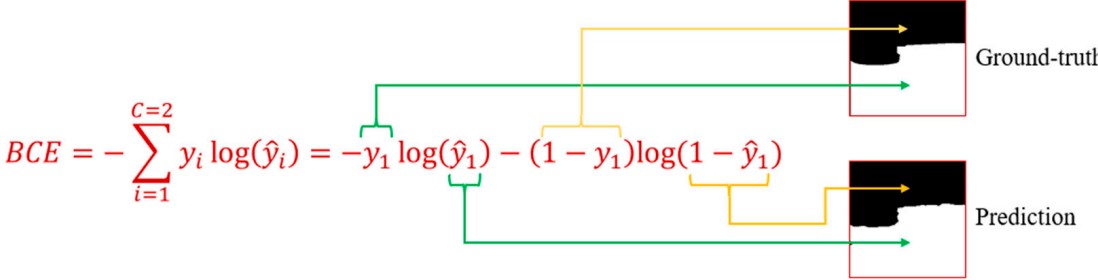

**Figure 2.** Ground truth and prediction in the loss function (BCE).

Network parameters are optimized for binary cross-entropy via one of the SGD variants' parameter learning strategies. In contrast to the suggestion made by Yang et al. [28], our network's capabilities were increased through data augmentation, in which the original versions of the training images are used. The training process is sped significantly thanks to the extensive use of transfer learning. The authors use trained VGG models to set the network parameters. The only challenging part is picking the right training environment. After training and validating many different network versions, as shown in [7], the authors settled on the following parameters: the momentum is 0.9, and the learning rate is 0.001; the training procedure is tuned for 300 epochs, and the learning rate is decayed by 10 for every 80 epochs.

When using a multi-scale FCN for binary semantic segmentation, the authors get two classes, accessible and inaccessible areas. Hence, with limited resources, enough performance and speed rate in data processing is necessary. The image size of $96 \times 96$ was exclusively chosen. Training times are cut thanks to the widespread use of transfer learning drastically. The authors use pre-trained VGG models to set up the network, with the most complicated part being the choice of training parameters through training and testing numerous network versions in Figure 3. Figure 3a represents the validation mIoU as the noise impacting the quality of the training process and how it can be reduced to speed up the training process. As a result, the accuracy of the forecast will drop. The authors take measures to avoid overfitting by augmenting their data. The method makes it easier to fine-tune the forecast shown in Figure 3b. Additionally, practically eliminating noise will considerably improve the quality of the prediction. The segmentation noise filtering method is finalized using the probabilistic models described in [28].

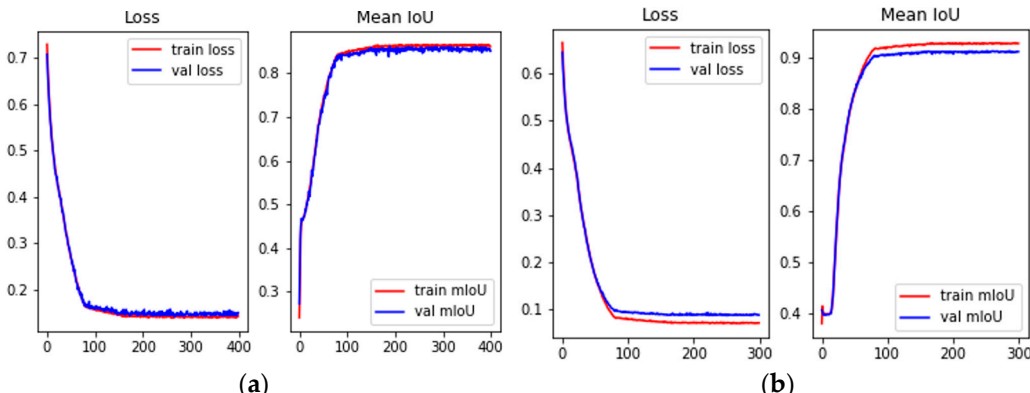

**Figure 3.** Training and validation on training images using mean intersection over union—mIoU as metric: (**a**) VGG-FCNs-CMU-96 × 96; (**b**) VGG-FCNs-Augment-96 × 96.

In these difficult scenarios, our network is able to accurately anticipate the ground border, demonstrating its tolerance to different corridor types. The segmentation experiment outcomes in the Ta Quang Buu library setting are depicted in Figure 4.

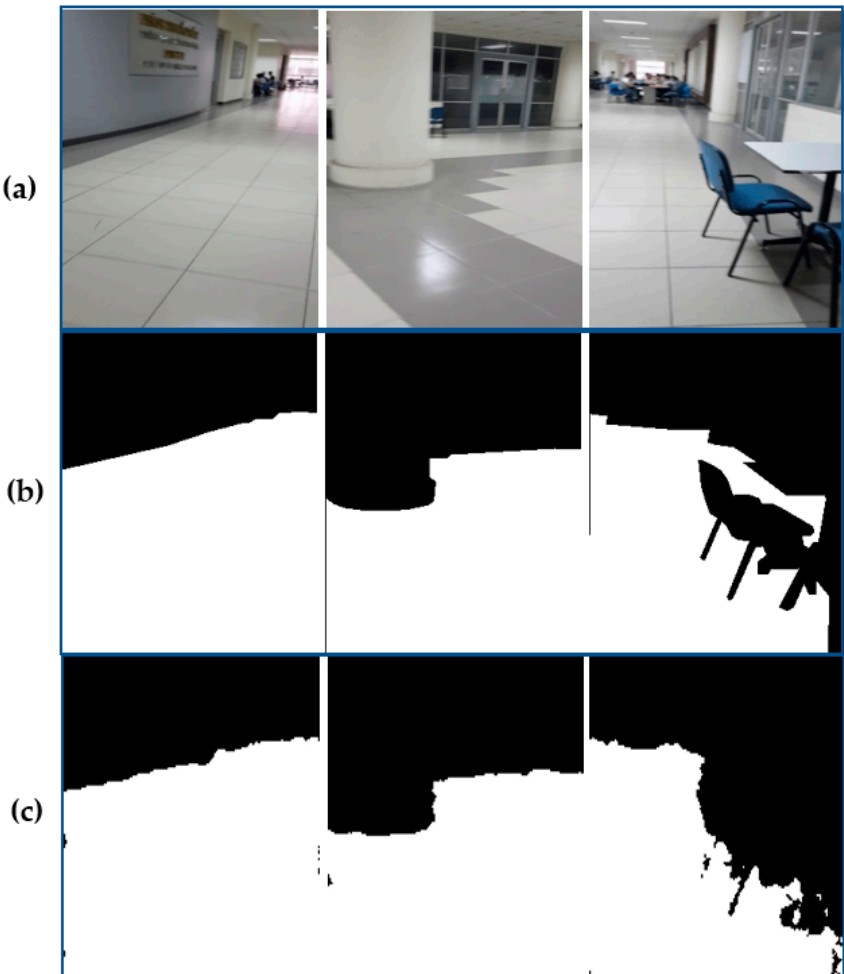

**Figure 4.** Segmentation results of the Ta Quang Buu library's environment: (**a**) raw images; (**b**) ground truths; and (**c**) predictions.

Finally, the optimal autonomous mobile robot path planning in the next section is guaranteed by the binary semantic segmentation VGG-16's output results.

## 3. Navigation Strategy Based on Ground Plane Segmentation

Our proposed approach constructs the fraction map based on the current observation, using the result of the binary semantic segmentation process in which the available area for movement of the arbitrary view will be labeled. Because inaccurate segmentation affects obstacle avoidance results. So, the segmentation model output will be continuously put into the probabilistic model using conditional random fields to filter the noise and uncertainty [29] completely. Furthermore, the quality of the architecture of the segmentation model also affects the results [30]. The navigation strategy of mobile robots includes three sections as follows: In Section 3.1, a perspective correction method is used to alter the ground region extracted from the semantic segmentation result. Section 3.2 provides a sequence of coordinates for precise navigation from the current place to the designated destination, based on a bird's-eye perspective. Finally, Section 3.3 concludes by discussing the ideal trajectory strategy for mobile robots. These three sections are further upon in the sections that follow.

### 3.1. Perspective Correction

Figure 5 shows the images obtained after the binary semantic segmentation procedure, labeling functional movement (white) and prohibited regions (black). Measurements of

the distance to the obstruction were complicated by the strong distortion caused by the perspective of the taken images.

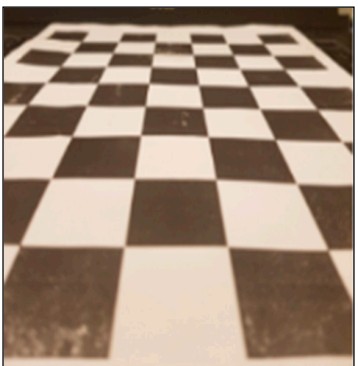 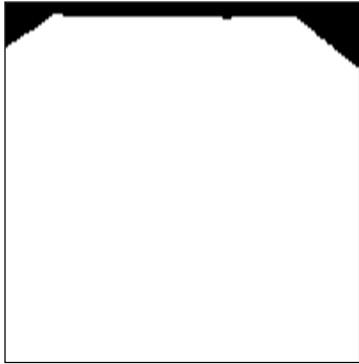

**Figure 5.** Result of semantic segmentation process.

The authors require coordination to be proportional to the actual ground in order to plan the trajectory. To get a rough idea of the connection between world coordinates and image coordinates, we can use a basic matrix multiplication based on the pinhole model and a mathematical camera description. The transformation that maps a point from the world coordinate system to the image coordinate system can be expressed as follows, where p and P denote the image point and the world point, respectively:

$$p = M_{int} \times M_{ext} \times P \tag{2}$$

where $M_{int}$ presents the matrix of intrinsic parameters, $M_{ext}$ is the matrix of extrinsic parameters. The intrinsic and extrinsic parameters are internal to the camera; however, the latter can change depending on the camera's position in the world frame.

The intrinsic and rotational constituents of the extrinsic parameters remain constant, owing to the camera's configuration mounted on the bird's eye view of the mobile robot. These parameters are presumed to remain invariant during movement, rendering the approximations calculable only once. Utilizing the correspondence of four points in [31], the front view of the ground plane can be constructed through a transformation matrix. Consequently, our resulting image solely captures the ground plane and the available and unavailable regions. The projective transformation, founded upon the four points, can be described as follows:

Let (x, y) and (x′, y′) be inhomogeneous. Coordinates of a pair of matching points x and x′ are in the world and image plane, and the given n point correspondences satisfy $(x, y) \leftrightarrow (x', y')$. Then, the transformation H is described such as follows:

$$x' = Hx_i \tag{3}$$

with each point correspondence satisfies two constraints:

$$x' = \frac{x_1'}{x_3'} = \frac{h_{11}x + h_{12}y + h_{13}}{h_{31}x + h_{32}y + h_{33}}; \; y = \frac{x_2'}{x_3'} = \frac{h_{21}x + h_{22}y + h_{23}}{h_{31}x + h_{32}y + h_{33}}. \tag{4}$$

Hence, H is determined uniquely. Hence, from any four points on the scene plane in Figure 6a, we will obtain four points of arbitrary view in Figure 6b. Finally, transformation H will build any four points of an arbitrary view of a real-time moving environment to any other four points in the frontal view.

The authors perform the perspective correction of the frontal views by using the checkerboard in Figure 7.

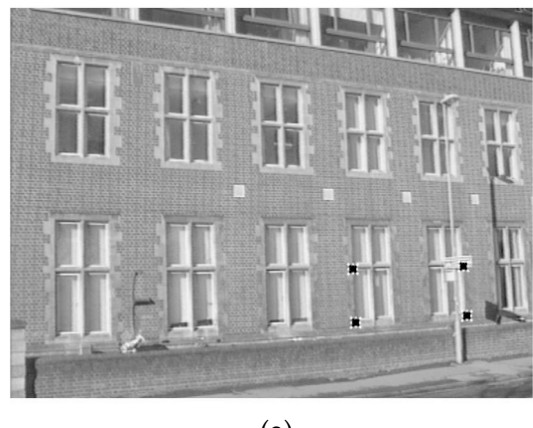
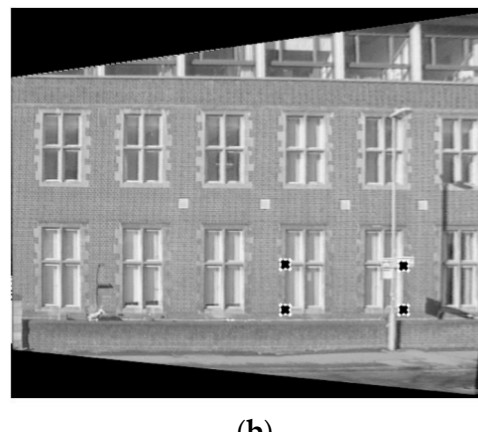

(**a**)                                                                              (**b**)

**Figure 6.** A projective transformation is defined by four points on the scene plane: (**a**) any four points arbitrary view of an environment's image, and (**b**) four points in the frontal view after using a projective transformation.

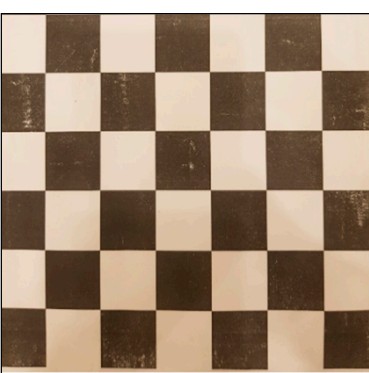
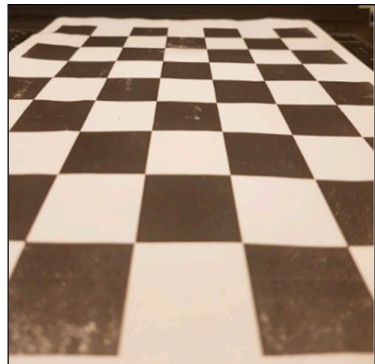

**Figure 7.** The checker board and the frontal view.

Using the above transformation, the authors can construct the frontal view for a real-time motion to define the available area. Figure 8 shows the segmentation model's output from frontal view images.

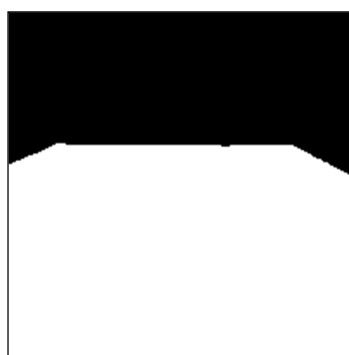
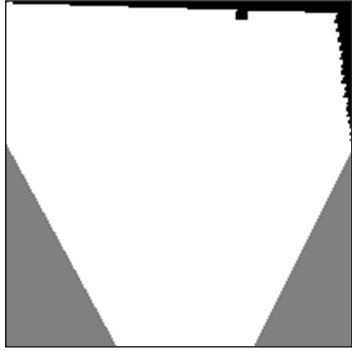

**Figure 8.** Source image and constructed frontal plane.

Based on the information perception of the monocular camera bird's view in Figure 9, mobile robot path planning will be designed entirely.

### 3.2. Path Planning

Based on the generated frontal plane, the authors will construct a moving path using a traditional A* algorithm [17]. Firstly, the authors create a collision-free area by scalding up the unavailable place. After that, the image will be divided into grid cells. The current

position is determined at the bottom center of the picture. The authors calculate the centers of each cell and use them for the path planning process in Figure 10.

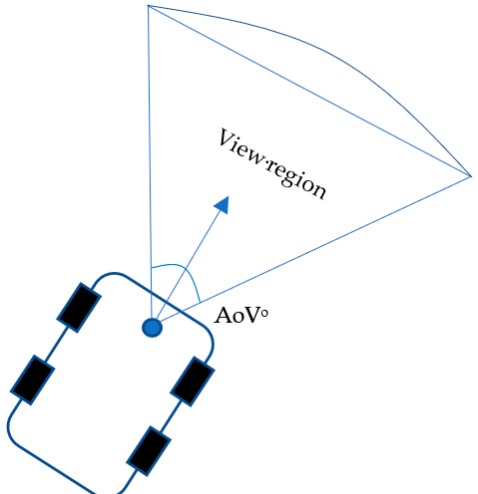

**Figure 9.** The mobile robot's bird view.

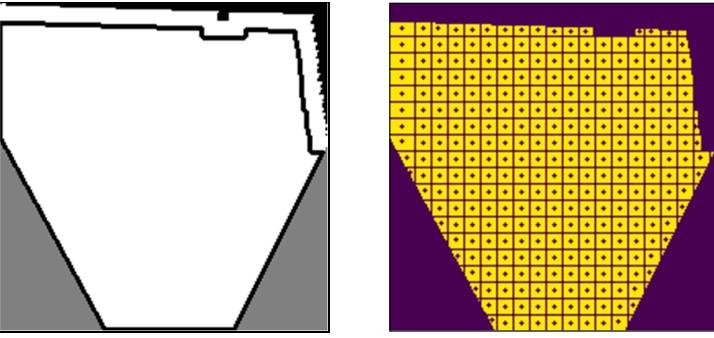

**Figure 10.** Detect collision-free area and cells division.

In each mobile robot position, we have five functional movements: left, up—left, straight, up—right, and right. A fundamental aspect of the A* algorithm is its utilization of a heuristic function to estimate the cost to the objective. The heuristic function is a distance-based mathematical estimation of the cost to the target from a given point. By utilizing a heuristic function, the A* algorithm can prioritize points that are likely to be closer to the target, enabling it to identify the shortest path more quickly in Equation (5).

$$f(m) = h(m) + g(m) \tag{5}$$

where m represents the current point, f(m) represents the cost evaluation function, h(m) represents the predicted cost from m to G, and g(m) represents the actual cost from m to the next point. The typical heuristic function for distance is the Manhattan distance, as indicated in Equation (6), or the Euclidean geometric distance, as given in Equation (7).

$$h_M(m) = |x_G - x_b| + |y_G - y_b| \tag{6}$$

and

$$h_E(m) = \sqrt{(x_G - x_b)^2 + (y_G - y_b)^2} \tag{7}$$

$(x_G, y_G)$ is the position of the goal point, and $(x_b, y_b)$ is the coordinate of any point.

Figure 11a shows that the A* algorithm can only search in four directions for neighborhoods when utilizing the Manhattan distance. Figure 11b presents an alternative use of Euclidean distance to study eight neighborhoods. The search process begins at S and then

expands to surrounding points. Then, the generation values following are calculated. The generating values are then evaluated using the heuristic function. Finally, the point with the smallest generation value is selected as the next parent point. The search procedure will be repeated until the target point G is located. In a large-scale map, the conventional A* search will generate an enormous number of path points and consume a significant amount of memory.

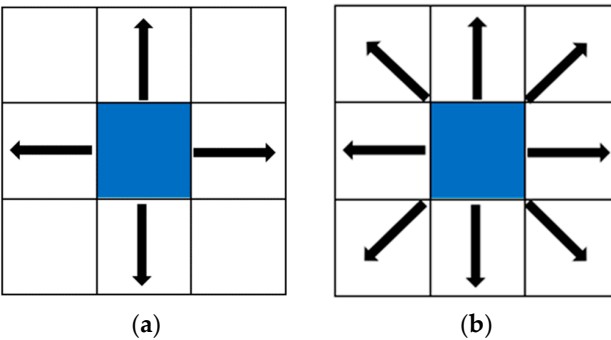

|   |   |
|:-:|:-:|
| (**a**) | (**b**) |

**Figure 11.** Symmetric path search: (**a**) four search directions of neighborhood; (**b**) eight search directions of neighborhood.

The heuristic cost is determined by the distance of the current grid cell's center to the target grid cell's center, and the path cost is calculated by the distance between the centers of adjacent grid cells. The algorithm repeats the exploration by picking the cell with the lowest price (sum of path cost and heuristic cost) in the priority queue until the picking cell reaches the target (path is found) or the queue is empty (path is not found). If the path is found, the algorithm will construct the absolute path by following the path cost from the target grid cell.

*3.3. Trajectory Optimization*

After obtaining the path by the A* algorithm, the authors use K points to describe the trajectories and apply the cost function (movement requirement) for the smoothening process. The cost function is constructed based on the necessity of a low steering angle, and a collision-free and continuous path, which is similar to the objective function described as follows:

$$\cos t_{total} = \omega_1 C_{collision} + \omega_2 C_{path} + \omega_3 C_{smooth} \tag{8}$$

where $\omega_1$, $\omega_2$, and $\omega_3$ are weights of collision cost, path cost, and smooth cost, respectively. Each cost component has a different effect on the result and can be described as follows:

Collision cost $C_{collision}$ prevents collision by penalty points that are in the unavailable area and pushes them to the nearest available area:

$$C_{collision} = \max(0, \text{bound val} - \text{distance}(p, p_b)) \tag{9}$$

where p are the path coordinates, $p_b$ are the closest collision-free bounding points relative to coordinates p, and bound val is the minimum distance to the nearest obstacle.

Path cost $C_{path}$ ensures the continuity of the sequence of coordinates by penalty points goes far from the planned path:

$$C_{path} = \text{distance}(\text{points}, \text{path points}) \tag{10}$$

where points are the coordinates, path points are the closest points in the planned path relative to points.

Smooth cost $C_{smooth}$ reduces the steering angle by penalty the sum of the distance of three consecutive points:

$$C_{smooth} = \text{distance}(p_i, p_{i+1}) + \text{distance}(p_i, p_{i-1}) \tag{11}$$

where $p_{i-1}$, $p_i$, $p_{i+1}$ are three consecutive coordinates.

The authors update the optimization using gradient-based methods [32]. In Equation (8), smooth cost $C_{smooth}$ introduces the risk of inaccuracy. The smooth cost function always decreased the angle between three adjacent coordinates. Thus, the authors apply a weight to the smooth cost based on whether the coordinates fall inside the accessible or unavailable area. When coordinates enter the available area, the update can typically continue. Otherwise, the magnitude of the update will be diminished by the added weight. The influence of the smooth cost remains, but it is less significant than the effect of the collision cost, which will return the coordinates to the available area.

## 4. Experimental Results

### 4.1. Proposed Obstacle Avoidance Strategy

All steps of obstacle avoidance and planning strategy are according to the optimization trajectory workflow in Figure 12 as below.

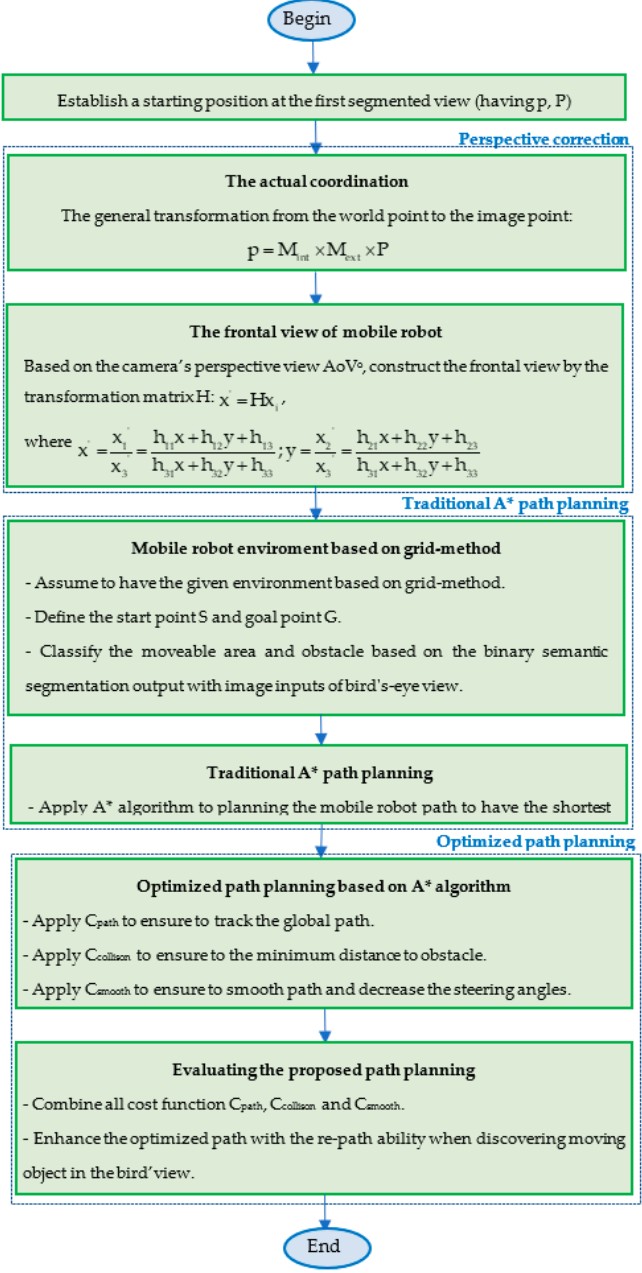

**Figure 12.** The optimization trajectory workflow.

### 4.2. Perspective Correction Process

In order to evaluate the created frontal plane, the authors will conduct studies on a checkerboard, which will provide patterns for improved recognition. In the initial experiment, the authors examine the non-obstacle region and assess the cell size in the generated image. The size of the cells in the created plane is proportionate to the size of the original cells in Figure 13. Hence, the cells in a constructed frontal plane can be utilized for qualitative measurement.

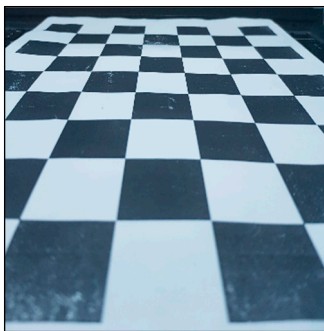 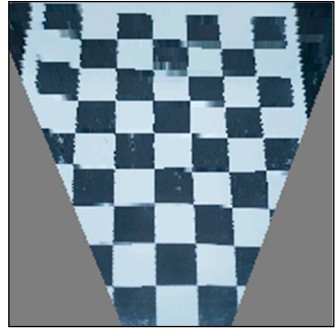

**Figure 13.** Perspective correction on checkerboard.

Then, the authors compare the differences between perspective correction with obstructions in various positions. Figure 14 depicts the relative position of an obstruction from an arbitrary and frontal perspective. The checkerboard cells can compare the near function while moving blocks in the checkerboard plane, observing that perspective correction is only performed to the checkerboard plane and causes an inaccuracy in the other plane.

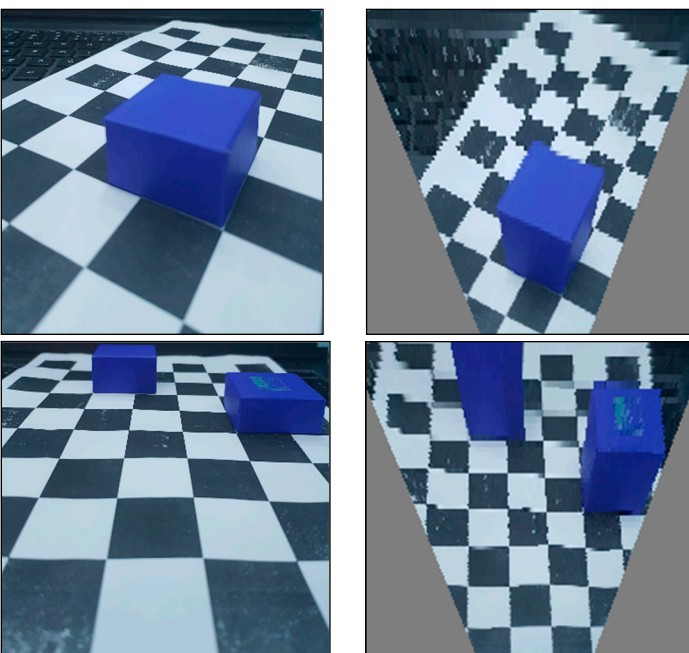

**Figure 14.** Relative obstacle position in frontal plane with four different obstacle positions.

Since the authors treat the unobservable area as an unavailable area for movement, the result of perspective correction can be used to determine four different obstacle positions in Figure 15.

### 4.3. Path Finding Process

Before implementing the path finding algorithm, the authors establish the collision-free region by considering the path as a sequence of points that locate the mobile robot's

center due to the collision problem. In Figure 16, the collision-free area is delineated by extending the unusable area (black area) by a distance equal to half the mobile robot's width. The variance in boundary dimension may result in a variable path planning outcome.

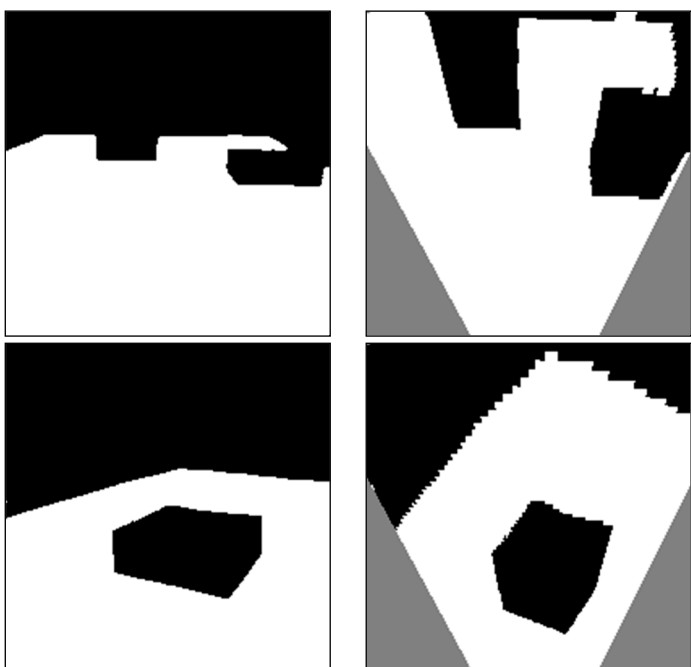

**Figure 15.** Perspective correction in semantic segmentation image with four different obstacle positions.

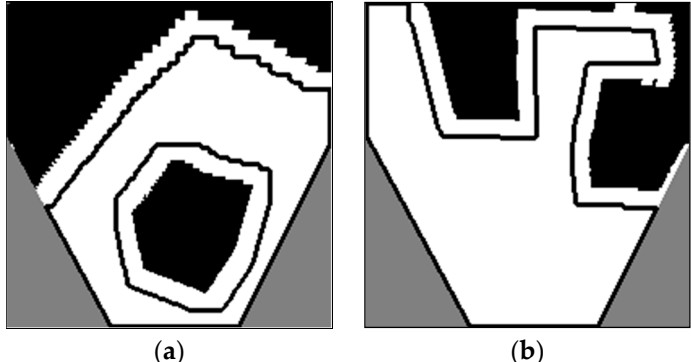

(**a**)                              (**b**)

**Figure 16.** Determine the bounding of the collision-free area: (**a**) having one obstacle at the bottom of the environment, (**b**) having two obstacles at the top of the environment.

The authors apply the A* path planning algorithm to the collision-free region. As the authors plan the path on the observable area, which was rather uncomplicated in Figure 17, the computational cost of the A* algorithm is modest.

### 4.4. Trajectory Optimization Process

In Figure 18, the authors examine the effect of the collision cost using random blue points and the same method. The empty black spot can be used to store the blue points. There are two possible scenarios in which a mobile robot encounters barriers while moving in real time: Figure 18a depicts the environment for Scenario 1, which features a single obstacle in the bottom center, and Figure 18b depicts the environment for Scenario 2, which features a pair of obstacles at the top. To prevent accidental collisions with obstacles, $C_{collision}$ changes blue points to black ones.

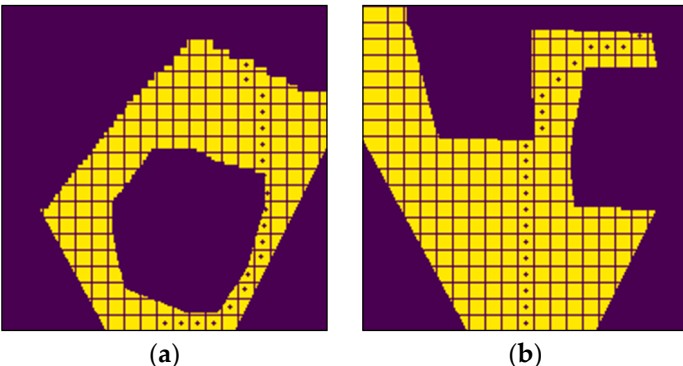

**Figure 17.** Path planning in the collision-free area in the conventional A* path: (**a**) having one obstacle at the bottom of the environment, (**b**) having two obstacles at the top of the environment.

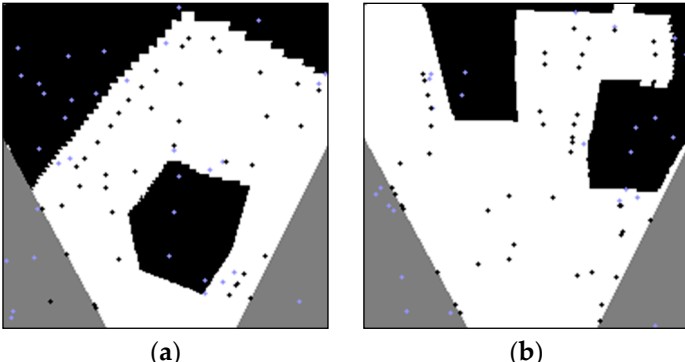

**Figure 18.** Effect of the collision cost $C_{collision}$ in the enhanced A* path: (**a**) having one obstacle at the bottom of the environment, (**b**) having two obstacles at the top of the environment.

The outcome demonstrates that path cost moves random blue points to the closest points along the intended path in Figure 19.

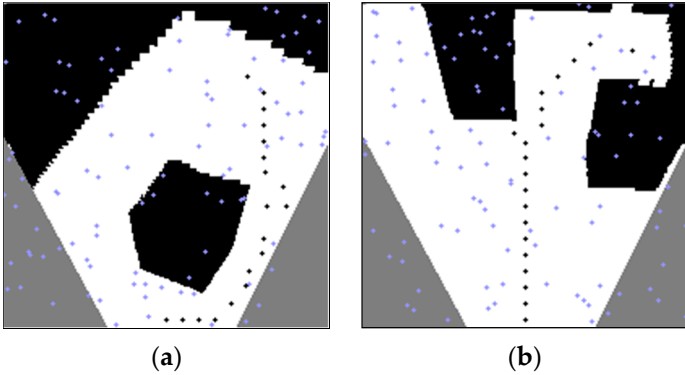

**Figure 19.** Effect of the path cost $C_{path}$ in the enhanced A* path: (**a**) having one obstacle at the bottom of the environment, (**b**) having two obstacles at the top of the environment.

In Figure 20, a comparison is presented between the performance of the original A* path and the path generated by combining the collision cost and smooth cost. To enhance the safety of the autonomous mobile robots (AMRs), the environment is modified to ensure the path remains unobstructed, as depicted by the blue line. Including the additional collision cost in the A* heuristic cost function is a remedy for the potential issue of robot collision (indicated by the black line) when traversing or turning around an obstacle. Application of the traditional A* algorithm may not be sufficient to avoid collisions in such scenarios. Furthermore, the smooth cost decreases the angle between three consecutive path points, leading to a smoother path.

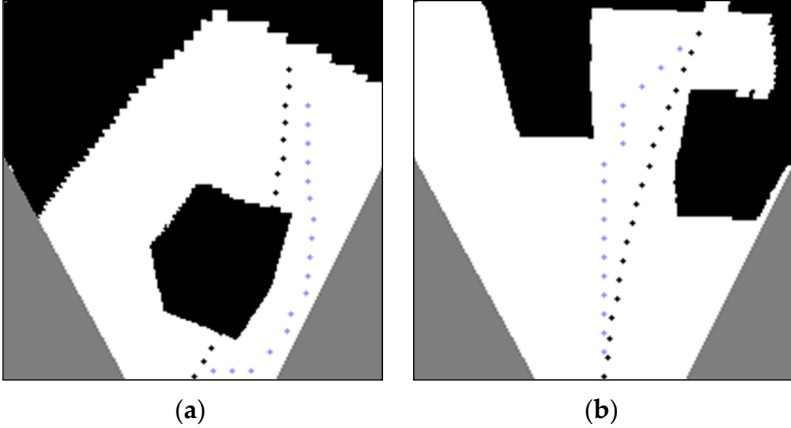

(**a**)                    (**b**)

**Figure 20.** Effect of the smooth cost in the enhanced A* path: (**a**) having one obstacle at the bottom of the environment, (**b**) having two obstacles at the top of the environment.

After separately studying path cost, collision cost, and smooth cost, we investigate all cost functions simultaneously and show the final results. The processing time of three scenarios is separately measured to prove the proposed path's performance in Figure 21.

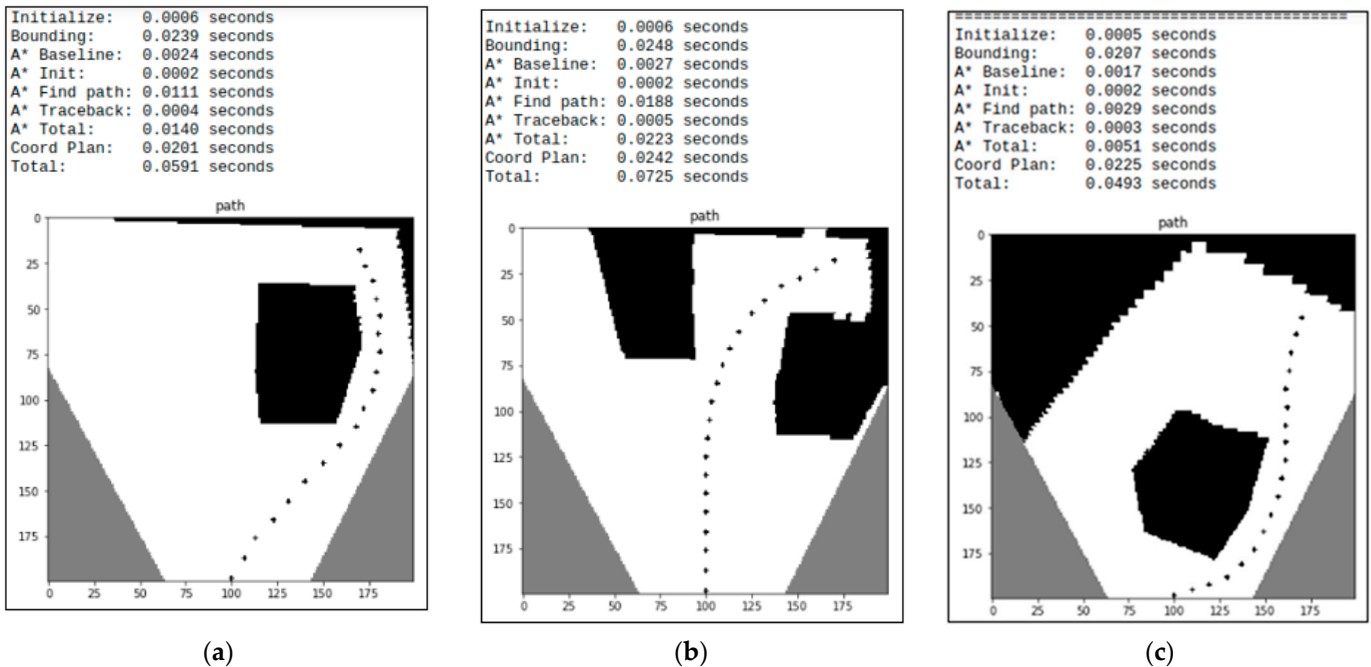

(**a**)                    (**b**)                    (**c**)

**Figure 21.** Final trajectory results with real-time measurement: (**a**) having one obstacle at the top right of the environment, (**b**) having two obstacles at the top of the environment, and (**c**) having one obstacle at the bottom of the environment.

More three situations tested in the simulation are as follows: Scenario 3 has one obstacle in the top right corner of the shifting environment, whereas Scenario 4 has two, and Scenario 5 has one massive obstacle at the base. To validate the quality of the segmentation model's design, the results are compared to those in [30]. The A* algorithm is first utilized to find the path based on optimized path planning. The overall completion time of the A* algorithm is only 0.0140, 0.0223, and 0.0051 s for Scenarios 1, 2, and 3, respectively. Finally, the enhanced A* algorithm with three cost functions, including path cost, collision cost, and smooth cost, directs the mobile robot to safely avoid obstacles and follow a smooth trajectory to its goal. In Scenario 3, the impressive processing time is 0.0591 s, in Scenario 4 it is 0.0725 s, and in Scenario 5 it is 0.0493 s.

With the successful application of the optimized mobile robot navigation strategy, particularly the process of smoothing the path while maintaining the mobile robot's safety, the tracking trajectory controlling is robust with steering angle variations of less than 0.2 rad in Figure 22.

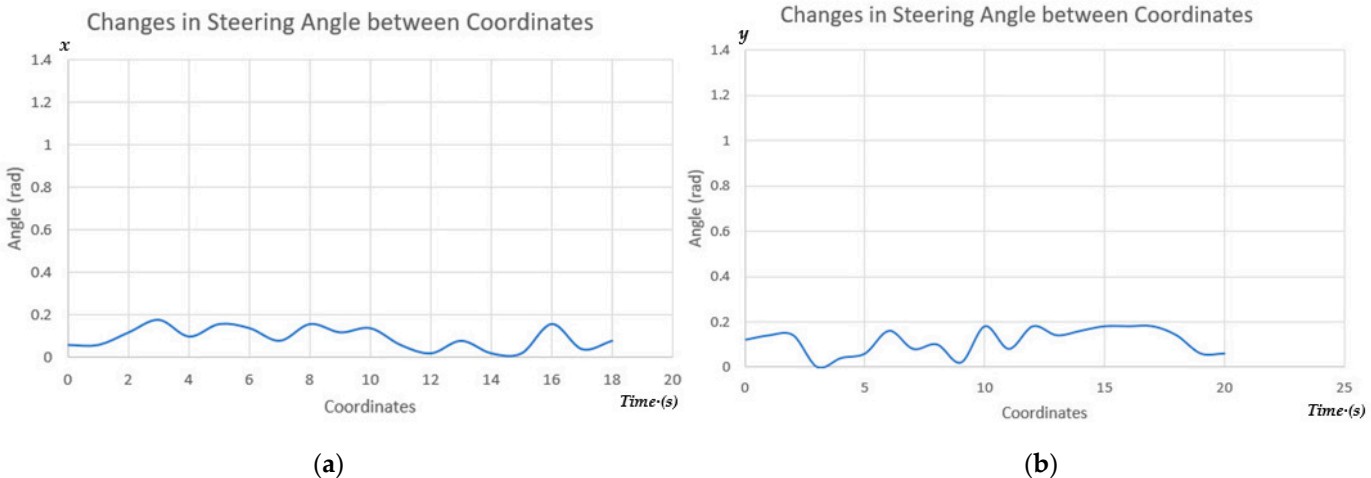

(**a**)                    (**b**)

**Figure 22.** Changes in steering angle while mobile robot is tracking optimal trajectory: (**a**) steering angle between coordinates in *x* axis, (**b**) steering angle between coordinates in *y* axis.

Furthermore, the authors continuously validate the mobile robot path planning performance in two continuous scenarios in Figure 23. The following 50 × 50 grid environments are continuously conducted. The start point S (0, 0) and goal point G (50, 50) were used as simulated test maps. The obstacles appeared randomly in Scenario 6, and the formation of obstacles is more complex in Scenario 7.

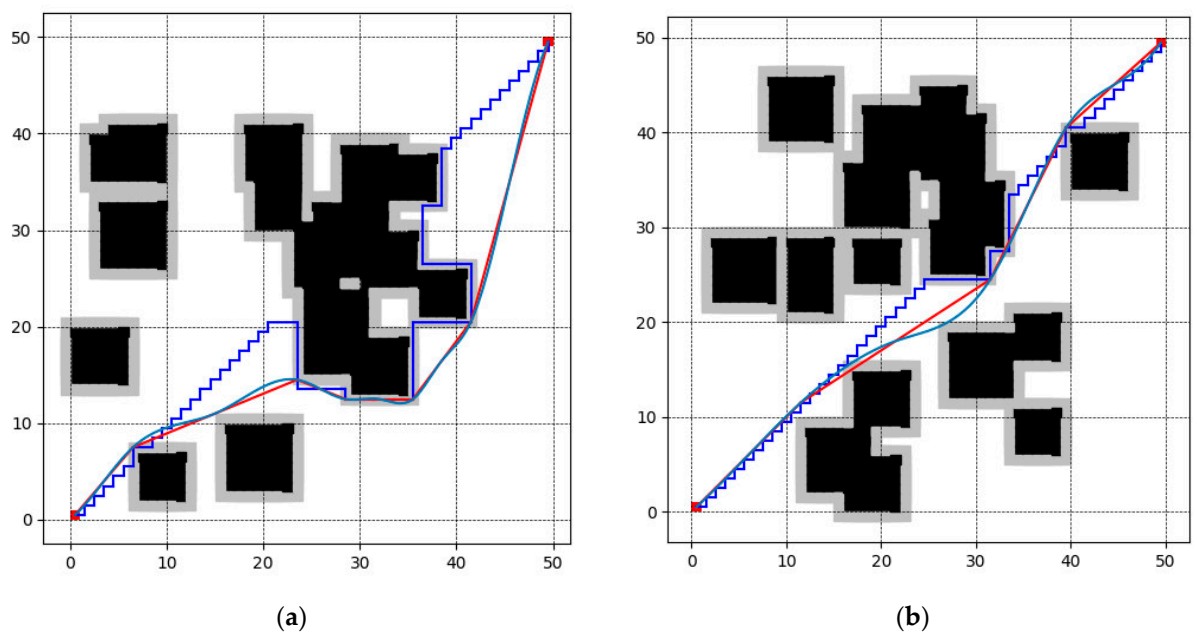

(**a**)                    (**b**)

**Figure 23.** Mobile robot path using traditional A* algorithm (blue lines), our improved A* algorithm without smooth cost (red lines), and our improved A* algorithm with all cost functions (green lines): (**a**) Scenario 6 with random obstacles and (**b**) Scenario 7 with more complex obstacles.

In both Scenarios 6 and 7, if only using A* path, the mobile robot will track only the global path and turn too many times. Hence, the A* path requires much more computational scale and memory usage. A* solution is not optimal compared with our improved A*

algorithm through the path length and time processing. Furthermore, A* path moves into the serious region around the obstacle defined by collision cost (grey region) in Figure 23. Based on the binary semantic segmentation image results, using path cost and collision cost, the performance of the mobile robot path is better (red lines), in the local area of the frontal view. The path is always a collision-free distance to obstacles. Finally, when adding more smooth costs, our improved A* makes the path smoother (green lines). The trajectory must be shorter than the planning path. Table 1 displays the results of a comparison between the A* algorithm and the suggested A*-based route planning algorithm in terms of the number of path nodes and the length of the paths taken in the aforementioned two cases.

**Table 1.** The comparison between our proposed path planning and different methods.

| Methods | Path Nodes | Path Length |
|---|---|---|
| Traditional A* algorithm [17] | 73 | 86.8 |
| Improved Dijkstra algorithm [16] | 52 | 80.5 |
| Our improved A* without smooth cost | 7 | 75.2 |
| Our proposed method | 7 | 75.6 |

The proposed method significantly reduced the calculation path nodes and path length in the mobile robot's movement compared to the traditional A* algorithm [17] and improved the Dijkstra algorithm [16]. Moreover, the mobile robot's trajectory is smoother and more robust when the changed steering angle is reduced in Figure 22. In addition, the proposed path planning still ensured the shortest distance between the start and goal points. The result also consolidates the optimal mobile robot navigation strategy design.

The authors will evaluate the results of the transformation and conclude that semantic segmentation is essential for constructing the frontal perspective of the ground. Then, the mobile robot's optimal path planning can be created. Practical experimental results improve collision-free zone detecting procedures. Optimized path planning with all of the cost functions and obstacle avoidance will be created once the front view becomes the norm. Figure 24 depicts the four-wheel mobile robot used to test our proposed semantic segmentation, while Figure 25 depicts the same robot navigating a 2.8 m 1.4 m area with four obstacles in Scenario 8. Figure 25 shows how our best obstacle avoidance method [7] replans a safe global path for the mobile robot to travel along, which it then uses to move forward.

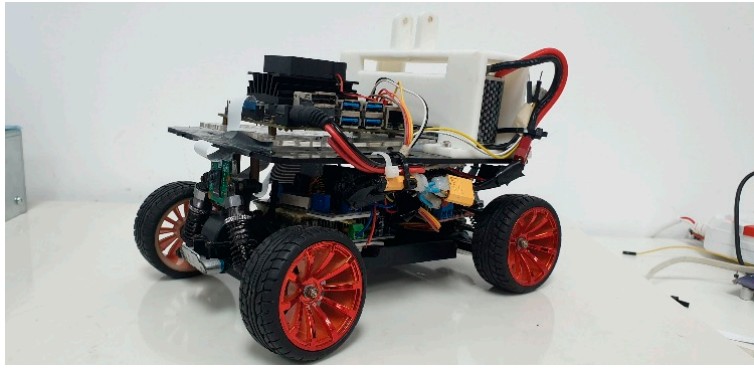 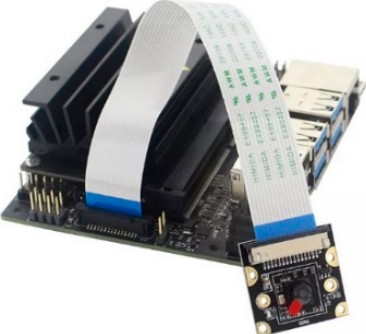

**Figure 24.** Jetson nano-powered mobile robot with 8-megapixel Raspberry Pi camera module.

To verify the moving obstacle avoidance, the following experiments are set up in Scenario 9 under ROS environment, in Figure 26: The authors postulate a mobile robot that follows a global path from point A to point B. The obstacle Obs moved from position Pos 1 to Pos 2 with a velocity as 0.2 m/s. The mobile robot's parameters were as follows: the maximum velocity: 1 m/s, the maximum angular velocity: $25°$/s, the an-

gular velocity resolution: $1°/s$, the acceleration: $0.25 \text{ m/s}^2$, and the angular acceleration: $45°/s^2$, respectively.

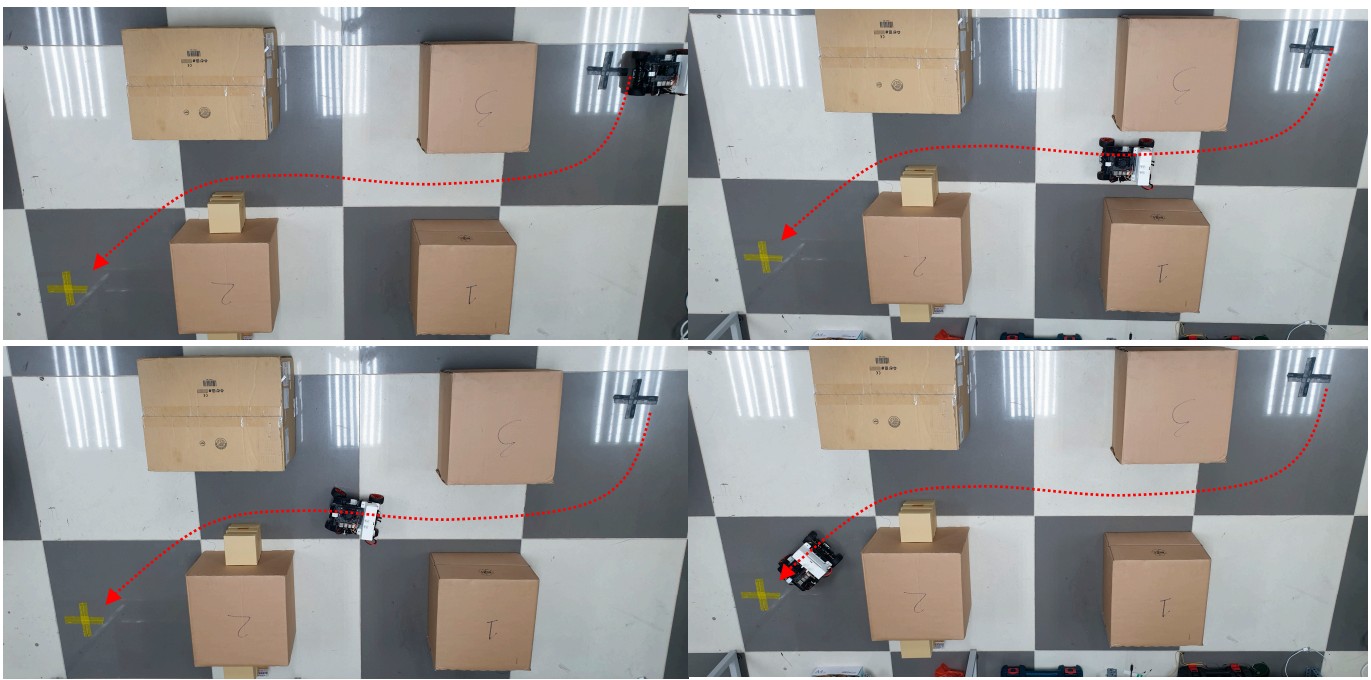

**Figure 25.** Mobile robot path in experimental four obstacles scenarios.

Figure 26 (snapshots (a)–(f)) depicts a mobile robot following a global path determined by the A* algorithm. Then, the mobile robot discovers the obstacle Obs, and the path is modified the local path to becoming the optimized path. Hence, the mobile robot avoids the obstacle successfully at Pos 1, then continues at the new Pos 2 of the moving obstacle. The experimental test is conducted on a mobile robot using the monocular camera in an actual ROS environment. In addition, the optimal mobile robot trajectory strategy ensures the mobile robot's robust movement while avoiding obstacles. All snapshots of Figure 25 present the mobile robot's moving process, including as follows:

In Figure 26a, at the start point S, a mobile robot will move to goal point G. The global path is entirely built. In each local area of the frontal view, the mobile robot discovers if it has obstacles in the path. Then, the mobile robot determines the door's position and adjusts the path to move through the door successfully. Next, the mobile robot movement maintains a safe collision distance from the obstacle Obs at Pos 1, as shown in Figure 26b. Furthermore, when the obstacle moves from Pos 1 to Pos 2, the way will intersect the first global path based on the A* algorithm. In Figure 26c, if the optimized mobile robot path is not successfully implemented, the robot will collide with the obstacle Obs in Figure 26b,c (red dotted lines). The mobile robot velocity is decreased from 1 m/s to a suitable velocity to gain enough time processing of implementing the optimized path. From Figure 26d–f, the mobile robot path is completely re-planned to the new path (green dotted lines). The mobile robot successfully tracks the optimized path from S to G with a collision-free distance to the obstacle of 0.2 m. Our evaluation findings demonstrate that the method successfully detects and avoids obstacles in real time with great precision and efficiency.

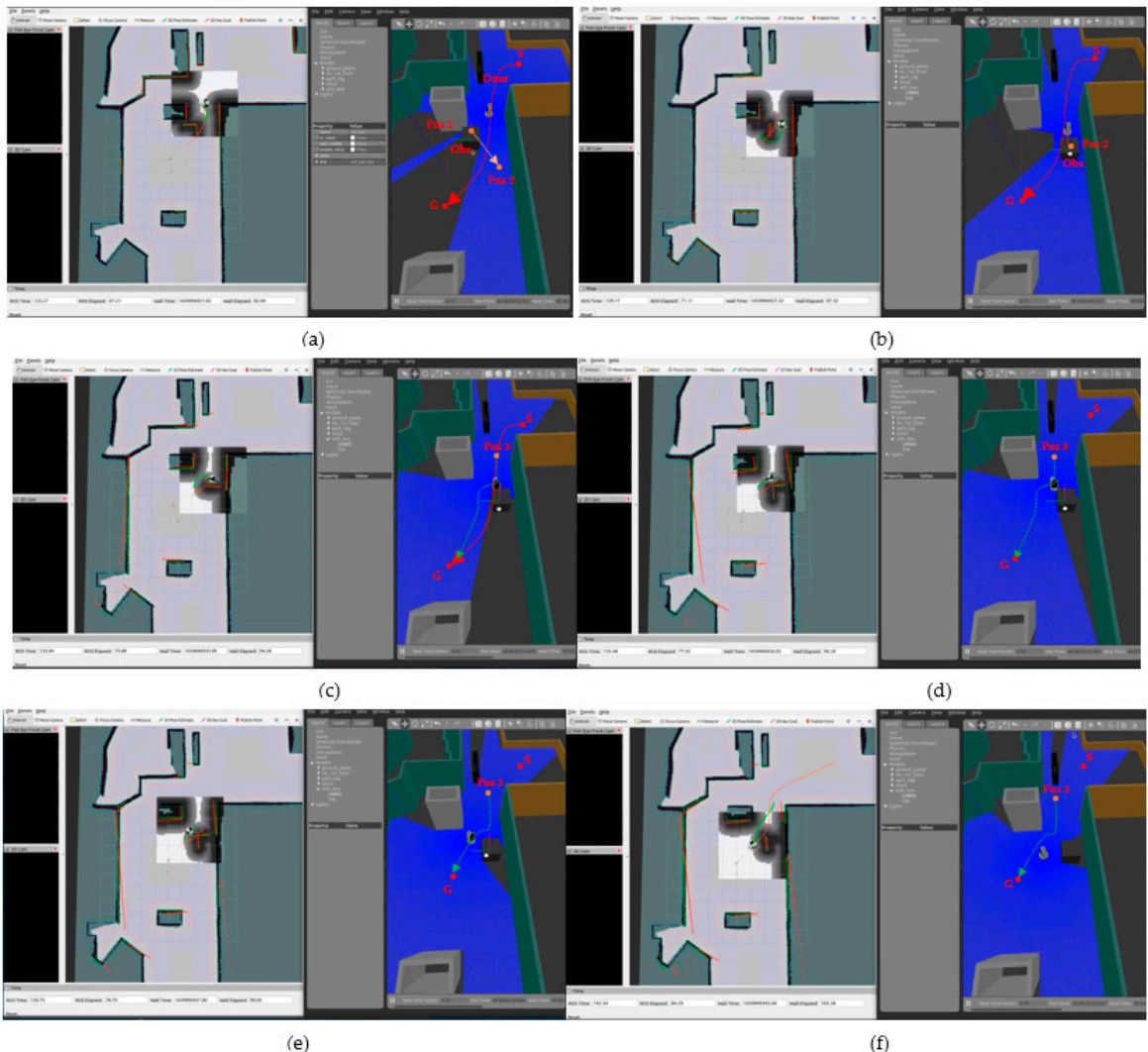

**Figure 26.** Optimized trajectory of mobile robot in the experimental moving obstacle scenario in ROS environment with the global path is dotted red lines and the re-path is dotted green lines: (**a**) Mobile robot moves from the start point S to goal point G, through the gate and avoid the obstacle Obs at Pos 1 (red dotted line); (**b**) After avoiding the obstacle Obs, mobile robot detect the moving obstacle at Pos 2; (**c**) The path is changed to ensure the moving obstacle Obs (green dotted line); (**d**) mobile robot moves successfully according to a new re-path; (**e**) mobile robot turn left to avoid the conner of the moving Obs; (**f**) mobile robot reaches to the goal point G with new re-path.

## 5. Conclusions

Path finding in the real world necessitates multi-objective optimization issues, such as optimizing the shortest path, the smallest distance to obstacles, and the smoothest trajectory. However, obstacles may arise when the qualities of the aims contradict. Thus, a multi-objective evolutionary algorithm based on binary semantic segmentation is proposed. Furthermore, data augmentation helps predict challenging situations, such as poor lighting conditions and background clutter. The most important parts of a good obstacle avoidance algorithm are collision-free path planning, a manageable amount of processing time, and minor adjustments to the steering angle. Instead of relying on the option of following an initial reference, as with existing systems, the mobile robots' navigation technique will be proactive in selecting flexible alternatives in the frontal view. Inputs for a perspective transform are collected from a segmentation map deemed suitable for perspective correction. From the beginning and ending locations of the map, the optimized path based on the A* algorithm will determine the direction of rapid movement.

Moreover, in the trajectory optimization process, with enhanced A* algorithm, the authors create a novel cost function construction based on the need for a path finding, collision-free path with a low steering angle. Collision cost $C_{collision}$ prevents collisions caused by penalty points in inaccessible regions and moves them to the nearest available region. Path cost $C_{path}$ ensures the continuity of the series of coordinates by assigning penalty points when the path deviates from its intended course. Finally, the expense is streamlined. $C_{smooth}$ decreases the steering angle by penalizing the distance between three consecutive points. After separately studying each component, the authors integrate the cost function and produce the final results. Authors measure the impressive processing time at each phase to evaluate the performance process and trajectory quality. A mobile robot can move stable and robots with the steering angle variations less than 0.2 rad. The simulation outcomes satisfactorily demonstrate the accuracy and enhancement of the proposed strategy. Our long-term objective is to train the model to distinguish between many different types of interior obstructions by using data from multiple classes. As a result, path planning will function better in a wide range of indoor environments. In addition, the gathered data could be used with sensor systems to deal with intricate issues in the global and local outside environment.

**Author Contributions:** Conceptualization, T.-V.D. and N.-T.B.; Project Administration, T.-V.D.; Supervision, N.-T.B.; Funding Acquisition: N.-T.B.; Methodology, T.-V.D. and N.-T.B.; Formal Analysis, N.-T.B.; Software, T.-V.D.; Experimental data and system, T.-V.D.; Writing—Review and Editing, T.-V.D. and N.-T.B. All authors have read and agreed to the published version of the manuscript.

**Funding:** This work was supported by the Centennial SIT Action for the 100th anniversary of Shibaura Institute of Technology entering the top 10 at the Asian Institute of Technology.

**Data Availability Statement:** The data that support the findings of this study are available on request from the corresponding author, Thai-Viet Dang.

**Acknowledgments:** The authors express grateful thankfulness to Vietnam-Japan International Institute for Science of Technology (VJIIST), School of Mechanical Engineering, HUST, Vietnam and Shibaura Institute of Technology, Japan.

**Conflicts of Interest:** The authors declare no conflict of interest.

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
