# Peer review of "Obstacle Avoidance Strategy for Mobile Robot Based on Monocular Camera"

_electronics, doi:10.3390/electronics12081932_

Round 1

Reviewer 1 Report

The Monocular method is really useful, and in this paper, it is presented with a good perspective and application, however, I think it should be improved in some aspects:
I don't know if it is relevant the use of the A algorithm, I think the contribution of the work is obstacle detection, but the planning is something beyond.
Beyond the processing time, the authors should present a comparable metric, a complexity analysis, or similar.
I consider that the experimentation scenario was very limited and that other cases should be presented.

I hope the comments will be useful to provide generalizations and improve the focus of the paper.

Author Response

Response letter for resubmission of the paper to the Electronics (MDPI)

Section: Computer Science & Engineering

Special Issue: Machine Learning Methods in Software Engineering

 Ph.D. Thai-Viet Dang

School of Mechanical Engineering.

Hanoi University of Science and Technology, Hanoi, Vietnam.

 [03/04/2023]

Dear Editor of the Electronics

Special Issue: Machine Learning Methods in Software Engineering

Thank you very much for allowing us to revise the manuscript “Obstacle Avoidance Strategy for Mobile Robot based on Monocular Camera.”  We want to thank the reviewers for their positive feedback and constructive comments. The reviewers’ suggestions and comments are beneficial to guide us in improving our manuscript.

We have addressed the reviewer’s concerns and believe the manuscript is significantly improved after making the suggested revision. Our detailed responses to the reviewer’s comments are added at the end of the revision. Changes in the revised manuscript are highlighted in color, corresponding to two reviewers’ comments. The revision has been carefully accomplished in consultation with all co-authors, and each author has approved the final version of this revision.

Thank you again for the consideration of our revised manuscript. We hope that the revised manuscript fulfills the requirements of the reviewers.

Best regards,

Thai-Viet Dang (On behalf of all authors)

Reviewer 2 Report

Generally, the subject of the paper is relevant and interesting.

The main problems and improvements of the paper:

·         The abstract does not include the paper motivations regarding the drawbacks and/or restrictions of the literature.

·         The introduction section does not include a brief description of mobile robots and the contributions of the paper.

·         All the elements and processes (or maybe only the processes in Subsection 4.1) should be illustrated with a workflow as an overall view of the work.

·         A* is a heuristic approach, so the solution is not always optimal. How is this limitation taken into account in the new method? Moreover, finding the path with A* should be described in the context with a pseudocode.

·         Based on the context, the robot captures images and the image processing is done in the Server. The power consumption for the data transferring between them should be clarified.

·         The bird's eye view should be briefly described in the context.

·         The performance of the new technique should be evaluated indifferent configurations, compared to the latest related work(s).

·         Abstract should address the main achievements of the paper based on a quantitative simulation summary.

·         The literature should be empowered by appending the other related works, especially the following works:

-          M. S. Gharajeh and H. B. Jond, “An Intelligent Approach for Autonomous Mobile Robots Path Planning Based on Adaptive Neuro-fuzzy Inference System,” Ain Shams Engineering Journal, vol. 13, no. 1, Article ID 101491, 2022.

-          M. S. Gharajeh and H. B. Jond, “Hybrid Global Positioning System-Adaptive Neuro-Fuzzy Inference System Based Autonomous Mobile Robot Navigation,” Robotics and Autonomous Systems, vol. 134, 2020.

·         The writing style requires major revision.

Author Response

Response letter for resubmission of the paper to the Electronics (MDPI)

Section: Computer Science & Engineering

Special Issue: Machine Learning Methods in Software Engineering

 Ph.D. Thai-Viet Dang

School of Mechanical Engineering.

Hanoi University of Science and Technology, Hanoi, Vietnam.

 [03/04/2023]

Dear Editor of the Electronics

Special Issue: Machine Learning Methods in Software Engineering

Thank you very much for allowing us to revise the manuscript “Obstacle Avoidance Strategy for Mobile Robot based on Monocular Camera.”  We want to thank the reviewers for their positive feedback and constructive comments. The reviewers’ suggestions and comments are beneficial to guide us in improving our manuscript.

We have addressed the reviewer’s concerns and believe the manuscript is significantly improved after making the suggested revision. Our detailed responses to the reviewer’s comments are added at the end of the revision. Changes in the revised manuscript are highlighted in color, corresponding to three reviewers’ comments. The revision has been carefully accomplished in consultation with all co-authors, and each author has approved the final version of this revision.

Thank you again for the consideration of our revised manuscript. We hope that the revised manuscript fulfills the requirements of the reviewers.

Best regards,

Thai-Viet Dang (On behalf of all authors)

Reviewer 3 Report

This works presents a a real-time obstacle avoidance strategy for mobile robots using a monocular camera using a neural network for a single camera and a well known A* planning method.

The paper has some relevant issues as follows:

-This reviewer believes there is not contribution to the state of the art to robotics community not the electronics community. If so, please clarify, see:

https://link.springer.com/article/10.1007/s00371-019-01714-6

-Experimental test with a physical robot testbed and camera are required for journal publication, only simulations are reported. Where is the robot?, what are its features? A video would be helpful.

-A comparison with state of the art schemes are inorder:

https://link.springer.com/article/10.1007/s11431-020-1582-8

-Some redaction and grammar issues are the following:

Line 90: "The Conclusion concludes...."

Author Response

(The authors gave the same response as above.)

Round 2

Reviewer 3 Report

Thank you for your work on all my comments, they were correctly addressed. I strongly suggest to put the response of my first comment directly on the paper. This is, to put the following in the manuscript:

1. The first main contribution of our paper is to design multi-scale fully convolutional networkbased semantic segmentation for mobile robot navigation using low resource system. The reference [7] proved the feasibility of proposed method for mobile robot using monocular camera in terms of computational cost and accuracy.

2. Another contribution is robust mobile robot navigation with small steering angle in tracking path planning. 

After this chance the paper now can be published.

Authors have correctly addressed all my comments, after puting the contributions in the paper, it can be published.

Author Response

Response letter for resubmission of the paper to the Electronics (MDPI)

Section: Computer Science & Engineering

Special Issue: Machine Learning Methods in Software Engineering

 Ph.D. Thai-Viet Dang

School of Mechanical Engineering.

Hanoi University of Science and Technology, Hanoi, Vietnam.

[14/04/2023]

Dear Editor of the Electronics

Special Issue: Machine Learning Methods in Software Engineering

Thank you very much for allowing us to revise the manuscript “Obstacle Avoidance Strategy for Mobile Robot based on Monocular Camera.”  We would like to express our thanks to the reviewers for their positive feedback and constructive comments about how to improve our manuscript.

We have addressed the reviewers' significant concerns and believe the manuscript is significantly improved after making the suggested revision. Our detailed responses to the reviewer's comments are added at the end of the respond letter. Changes in the revised manuscript are marked up using the “Track Changes” function in Word, corresponding to the reviewers’ comments. The revision has been carefully accomplished in consultation with all co-authors, and each author has approved the final version of this revision.

Thank you again for the consideration of our revised manuscript. We hope that the revised manuscript fulfills the requirements of the reviewers.

Best regards,

Thai-Viet Dang (On behalf of all authors)
